# Novel Biomarkers for Early Detection of Acute Kidney Injury and Prediction of Long-Term Kidney Function Decline after Partial Nephrectomy

**DOI:** 10.3390/biomedicines11041046

**Published:** 2023-03-28

**Authors:** Marco Allinovi, Francesco Sessa, Gianluca Villa, Andrea Cocci, Samantha Innocenti, Maria Zanazzi, Lorenzo Tofani, Laura Paparella, Dritan Curi, Calogero Lino Cirami, Riccardo Campi, Andrea Mari, Agostino Ognibene, Maria Lorubbio, Alessandra Fanelli, Stefano Romagnoli, Paola Romagnani, Andrea Minervini

**Affiliations:** 1Nephrology, Dialysis and Transplantation Unit, Careggi University Hospital, 50139 Florence, Italy; samantha.innocenti@unifi.it (S.I.); .; 2Unit of Oncologic Minimally-Invasive Urology and Andrology, Careggi Hospital, 50139 Florence, Italy; 3Department of Anaesthesia and Intensive Care, Careggi University Hospital, 50139 Florence, Italy; 4Department of Health Sciences, Section of Anaesthesiology, Intensive Care and Pain Medicine, University of Florence, 50139 Florence, Italy; 5Laboratory Department, Careggi University Hospital, 50139 Florence, Italy; 6Pediatric Nephrology Unit, Meyer Children’s Hospital, 50139 Florence, Italy; 7Excellence Centre for Research, Transfer and High Education for Development of DE NOVO Therapies (DENOTHE), University of Florence, 50139 Florence, Italy; 8Department of Experimental and Clinical Medicine, University of Florence, 50139 Florence, Italy

**Keywords:** acute kidney injury, biomarkers, NephroCheck, NGAL, partial nephrectomy, renal cell carcinoma, TIMP-2, IGFBP7

## Abstract

Background: Identifying acute kidney injury (AKI) within few hours of onset is certainly helpful. However, early prediction of a long-term eGFR decline may be an even more important goal. Our aim was to identify and compare serum [creatinine, kineticGFR, cystatin C, neutrophil gelatinase–associated lipocalin (NGAL)] and urinary (NephroCheck, NGAL, proteinuria, albuminuria, acantocytes at urinary sediment) predictors of AKI that might efficiently predict long-term GFR decline after robotic Nephron-Spearing Surgery (rNSS). Methods: Monocentric prospective observational study. Patients scheduled for rNSS for suspected localized Renal Cell Carcinoma from May 2017 to October 2017 were enrolled. Samples were collected preoperatively and postoperatively (timepoints: 4 h, 10 h, 24 h, 48 h), while kidney function was re-assessed up to 24 months. Results: 38 patients were included; 16 (42%) developed clinical AKI. The eGFR decline at 24 months was more pronounced after postoperative AKI (−20.75 vs. −7.20, *p* < 0.0001). KineticGFR at 4 h (*p* = 0.008) and NephroCheck at 10 h (*p* = 0.001) were, at multivariable linear regression analysis, efficient predictors of post-operative AKI and long-term eGFR decline if compared to creatinine (R2 0.33 vs. 0.04). Conclusions: NephroCheck and kineticGFR have emerged as promising noninvasive, accurate, and early biomarkers of postoperative AKI and long-term GFR decline after rNSS. Combining NephroCheck and kineticGFR in clinical practice would allow to identify high risk of postoperative AKI and long-term GFR decline as early as 10 h after surgery.

## 1. Introduction

Surgery is the gold standard for the treatment of localized Renal Cell Carcinoma (RCC). Although oncological outcomes are similar in radical nephrectomy (RN) and in nephron-sparing surgery (NSS), the incidence of hypertension and/or chronic kidney disease (CKD) in the long-term is significantly higher after RN [1,2]. CKD affects quality of life and has important health-economic implications being associated with increased risk of cardiovascular events and renal failure [3]. Similarly, patients with a lower estimated Glomerular Filtration Rate (eGFR) after renal surgery, experience significantly poorer survival [4], strongly supporting the innovative statement “every unit of glomerular filtration rate counts” [5], in order to improve long-term renal function. Current guidelines strongly favor partial nephrectomy (PN) for all patients with T1 tumors [6], and this indication is even stronger in elderly and comorbid patients with uncontrolled hypertension, diabetes, preoperative CKD stage 3 or greater, and/or proteinuria.

In addition, patients undergoing kidney surgery sometimes present with a pre-existing CKD at the time of surgery [2], and perioperative episode of acute kidney injury (AKI) may contribute to poor renal recovery. Post-operative AKI is a serious complication after kidney surgery, reported in up to 41% of patients undergoing PN, and its diagnosis is often delayed and likely underestimated [7,8]. In patients undergoing PN, AKI has been associated with long-term CKD [9], prolonged hospital stay, cardiovascular events, overall mortality [10,11], and tumor relapse [12].

New consensus criteria provided precise definitions of the severity of AKI [13,14]. However, they rely on serum creatinine (sCr), which is known to be an inaccurate marker of AKI as it is influenced by several non-renal factors, its increase is delayed from 12 to 48 h and potentially masked by hemodilution [15]. 

Therefore, several new urinary and serum biomarkers promise to address the gap with the use of sCr [8,16,17,18]. Novel biomarkers may complement sCr measurement or most likely improve the diagnostic accuracy of AKI, predicting development of CKD. The ability to detect AKI within few hours of onset would be helpful in protecting renal function. However, early prediction of a long-term eGFR decline may be an even more important goal.

The aim of this monocentric prospective observational study is to identify those serum and urinary markers of AKI that might efficiently predict long-term GFR decline after robotic NSS. Different biomarkers of AKI were selected in order to represent different sites of injury in the nephron and different mechanisms of kidney injury (Figure 1). Kinetic of these biomarkers within 24 h after surgery and their predictive capability in early identifying AKI are also explored.

## 2. Materials and Methods

We prospectively observed all consecutive patients scheduled for robotic NSS in suspected localized RCC at the Careggi University Hospital, Florence, Italy, between May 2017 and October 2017. Inclusion criteria were: (1) age > 18 years; (2) clinical evidence of a sporadic solid renal mass (T1, N0, M0); (3) at least 2 risk factors for long term GFR decline: age > 70 years, preoperative anemia, chronic heart failure (NYHA > 2), diabetes, preoperative CKD (defined according to KDIGO classification) [19]. Patients in whom at least two preoperative sCr values were not available were excluded from this study as well as patients who presented with a preoperative eGFR < 30 mL/min/1.73 m^2^ (Appendix A).

Collected data also included demographic, preoperative and postoperative clinical information, anesthesiologic and surgical variables.

The development of postoperative AKI (according to the Kidney Disease: Improving Global Outcomes [KDIGO] criteria) [14,19] was assessed. A sCr increase ≥ 0.3 mg/dL within 48 h or a sCr increase of 1.5 times the baseline value known to have occurred within the prior 7 days, or a urine (fresh samples, collected from the bladder catheter) output < 0.5 mL/kg/h for at least 6 h, were identified as clinical AKI. Blood and urine samples were collected preoperatively and at predefined time points postoperatively (4 h, 10 h, 24 h, 48 h), day-time and night-time.

The kineticGFR and the hourly urinary output, along with plasma levels of sCr, cystatin C, neutrophil gelatinase–associated lipocalin (NGAL), urine concentrations of protein, albumin, creatinine, NGAL, [TIMP-2] × [IGFBP7] (NephroCheck^®^ Test, San Diego, CA, USA), and acanthocytes in urinary sediment were analyzed (Figure 1). All these biomarkers were tested at different time points postoperatively against GFR long-term decline and AKI prediction. 

Patients’ kidney function was re-assessed up to 24 months postoperatively, to quantify the long-term eGFR decline and identify those biomarkers that might efficiently predict it (as primary outcome). Only those patients with complete evaluation of renal function at 12- and 24-month follow-up post-surgery were included (Appendix A).

The study was conducted in accordance with the Declaration of Helsinki; the study protocol was approved by local ethics committee (Comitato Etico Regionale per la Sperimentazione Clinica della Regione Toscana; study approval number BIO.16.015). Written informed consent was obtained from each subject involved in the study.

### Statistical Analysis

Results are expressed as median and interquartile range (IQR) or absolute and relative percentage when appropriate. The receiver operating characteristics (ROC) curve and univariate logistic regression analysis assessed the ability of biomarkers to predict eGFR decline and AKI. Odds ratios and ROC-AUC are expressed with their own 95% confidence intervals [95%CI]. Correlation between biomarkers levels postoperatively and the amount of GFR decline at 24 months was tested using Pearson correlation coefficient; ROC curve and univariate regression analysis assessed the ability of biomarkers to predict GFR decline at the same time point. Considering the eGFR decline as the primary outcome of this study, a predictive model was proposed using biomarkers associated with GFR decline with a *p*-value < 0.1 at univariate analysis. The model that identifies GFR decline most efficiently was tested against the predictive capability of sCr. The same model was tested for AKI prediction. Finally, to evaluate the association between different parameters with GFR decline at 24 months, a multiple linear regression model with backward selection method was performed including known predictive biomarkers and known risk factors for long-term eGFR decline as covariates. A *p* value < 0.05 was considered for statistical significance.

## 3. Results

Overall, 38 patients were included. Surgery was performed by a single experienced surgeon (>500 robotic PNs) and required arterial clamp in 23/38 (60.5%) cases with a median ischemia time of 20 (IQR 15–23) minutes. Overall, 16 (42%) patients developed a clinical AKI based on sCr and/or urine output (3, 9 and 4 patients based on urinary output criteria, sCr criteria and through both sCr and urinary output criteria, respectively). Among the remaining 22, about half of the patients had at least one biomarker of AKI increased postoperatively (subclinical AKI). Patients with clinical AKI were classified as stage 1 in 13/16 (81%) and as stage 2 in 3/16 (19%); no patient was classified as AKI stage 3.

Clinical, demographic, anesthesiologic and surgical variables overall and for patients who did and who did not experience AKI are reported in Table 1. Among different baseline parameters, preoperative serum NGAL was significantly higher in AKI patients (*p* = 0.002).

KineticGFR reached lowest value 4 h after surgery, proteinuria and albuminuria reached peak 4 h after surgery, urinary NephroCheck, serum and urinary NGAL increased 4 h and reached peak 10 h, while sCr and cystatin C increased 4 h and reached peak 24 h after surgery. Figure 2 and Appendix A describe the postoperative change in biomarker levels from baseline to 24 h.

A NephroCheck increase >0.3 (the previously validated cut-off value for AKI prediction) [20] was observed in 13/16 (81%) and 6/22 (27%) patients respectively for AKI and non-AKI groups at 4 h, and 14/16 (88%) and 11/22 (50%) at 10 h. Considering the highest NephroCheck value observed within the first 24 h postoperatively, an OR of 33.0 [1.55–700.16] was calculated for NephroCheck in predicting AKI (*p* = 0.025), ROC-AUC 0.75 [CI95% 0.64–0.86].

A serum Cystatin C increase ≥10% from the baseline within 24 h (a cut-off value previously validated for AKI prediction) [21,22] was observed in 8/16 (50%) and 1/22 (5%) patients respectively for AKI and non-AKI groups at 10 h. Considering the highest Cystatin C value observed within the first 24 h postoperatively, an OR of 148.11 [13.19–1662.9] was calculated for Cystatin C in predicting AKI (*p* < 0.0001), ROC-AUC 0.95 [CI95% 0.87–1.00].

Although not validated [23], a kineticGFR reduction ≥50% from the baseline GFR (the cut-off value previously validated for AKI definition in RIFLE criteria) [14] was observed in 9/16 (56%) and 4/22 (18%) patients respectively for AKI and non-AKI groups at 4 h. Considering the lowest value observed within the first 10 h postoperatively, an OR of 6.64 [CI95% 1.54–28.66] was calculated for kineticGFR in predicting AKI (*p* = 0.011), ROC-AUC 0.72 [CI95% 0.57–0.87].

Among different biomarkers, kineticGFR and serum Cystatin C showed the highest specificity in predicting an AKI episode, while NephroCheck showed the highest sensitivity (Table 2).

Postoperative serum and urinary NGAL, acanthocytes in urinary sediment and albuminuria were not significant predictive factors neither for prediction of AKI nor for long-term GFR decline after PN.

The median eGFR decline was −4.9 mL/min at 6 months, −8.0 mL/min at 12 months, −8.5 mL/min at 18 months, the decline at 24 months after surgery was −11.5 mL/min, more pronounced in those patients who developed AKI postoperatively (−20.75 vs. −7.20 mL/min, *p* < 0.0001) (Figure 3).

Among 8 patients without clinical AKI and without subclinical AKI (no kineticGFR reduction ≥50% and no NephroCheck increase >0.3), the median eGFR decline was −1.8 (IQR −4.5; 5.1) mL/min at 24 months after surgery, not significantly different from the eGFR decline of healthy controls.

A multiple linear regression model with backward selection method was performed to evaluate the association between different parameters with eGFR decline at 24 months. Among different covariates, NephroCheck at 10 h (*p* < 0.001) and kineticGFR at 4 h (*p* < 0.01) were the only independent factors predicting eGFR decline at 24 months.

Although several biomarkers were able to detect postoperative AKI (Appendix A), only uNGAL, NephroCheck, and kineticGFR predicted eGFR decline over a long-term follow-up (Appendix A). Correlation coefficients between biomarkers of postoperative AKI and long-term GFR decline are shown in Appendix A.

KineticGFR at 4 h (*p* = 0.008) and urinary NephroCheck at 10 h (*p* = 0.001) efficiently predicted eGFR decline at 24 months postoperatively (R^2^ = 0.33). These biomarkers outperformed postoperative sCr (R^2^ 0.33 vs. 0.04).

The same model was tested for AKI prediction. The kineticGFR at 4 h and urinary NephroCheck at 10 h could predict postoperative AKI similarly to sCr (AUC 0.86 [0.75; 0.98] vs. 0.83 [0.70; 0.97], *p* = 0.66), but 12 to 24 h earlier.

## 4. Discussion

Identification of patients that are more susceptible to AKI and CKD development after conservative treatment for renal tumors is of paramount importance [10,11], as these patients would be amenable of a tight postoperative surveillance and promptly referred to nephrology evaluation [24], in order to recognize early kidney damage, promote pharmacologic and non-pharmacologic strategies for nephroprotection [25], and reduce the progression to CKD and ESRD.

In this monocentric, prospective, observational study, the urinary NephroCheck measured at 10 h and kineticGFR at 4 h postoperatively have been demonstrated more efficient than sCr in predicting long-term eGFR decline and more prompt in predicting AKI after robotic NSS.

Considering the important limitations of sCr [15], a multiparametric approach to detect the postoperative kidney insult (both impairment and dysfunction) should be adopted [16]. Biomarkers of AKI might inform earlier on postoperative AKI development and more comprehensively on pathophysiology of kidney insult.

A recent systematic review [7] pointed out the potential use of biomarkers, such as serum cystatin C and urinary NGAL, as promising and early tools to predict AKI after PN or RN. Nevertheless, most of the selected studies in this systematic review adopted a not validated AKI definition, and postoperative urinary output assessment was frequently lacking. Moreover, long-term kidney function after NSS is neglected in most of these studies, as well as a comprehensive comparative analysis among different biomarkers of AKI (often limited to few examples for each study). Furthermore, other promising biomarkers of kidney dysfunction have never been tested in NSS, e.g., NephroCheck test^®^, acanthocytes in urinary sediment or kineticGFR.

Our study explored a comprehensive panel of 8 biomarkers and predictors (Figure 1) for early AKI and long-term GFR decline after robotic PN, including, for the first time, NephroCheck^®^ test, acanthocytes in urinary sediment and kineticGFR. Although most of the biomarkers of kidney damage have been associated with clinical AKI in this study (often with a more prompt time profile compared to sCr) (Appendix A; Figure 2), only few of them have been also associated with a clinically important long-term eGFR decline (Table 2). For these reasons, and considering limited resource usually available for perioperative medicine, in this study we aim at identifying those biomarkers that, compared to standard practice (i.e., postoperative sCr assessment), might be helpful for predicting AKI and long-term GFR decline earlier and more efficiently.

In patients who underwent robotic PN, the median eGFR decline was −11.5 mL/min at 24 months after surgery, with a significantly more pronounced GFR decline in those patients who developed AKI postoperatively (−20.75 vs. −7.20, *p* < 0.0001) (Figure 3).

Postoperative sCr was inaccurate as marker for prediction of long-term eGFR decline in this cohort. In particular, in our study, sCr at 24 h was correlated with eGFR decline at 24 months with a CC of −0.25 (*p* = 0.119). Interestingly, only 14/21 (66.6%) of patients with a significant eGFR decline (e.g., higher than 10 mL/min at 24 months) had experienced postoperative AKI. Interestingly, 6/7 (86%) of these patients were biomarkers positive (NephroCheck and serum NGAL), suggesting that subclinical AKI was responsible for postoperative kidney insult and long-term GFR decline. A significant eGFR decline occurred in only 1/21 (5%) patients without a positive biomarker, confirming their high sensitivity for long term renal prognosis.

Considering the 8 tested biomarkers of AKI, only uNGAL, NephroCheck, proteinuria and kineticGFR significantly predicted eGFR decline at 24 months (Table 2).

The value of proteinuria in predicting long-term eGFR decline has been previously demonstrated, even in patients undergoing PN or RN [26]. We identified for the first time a predictive capacity of proteinuria at 10 h postoperatively, as expression of both preoperative proteinuria (chronic damage) and new-onset proteinuria (acute injury and low renal functional reserve).

Among tested biomarkers, kineticGFR at 4 h (CC 0.34 [0.02; 0.59]) and urinary NephroCheck at 10 h (CC −0.47 [−0.69; −0.18]) predicted eGFR decline at 24 months most efficiently. Notably, if used together, kineticGFR at 4 h and urinary NephroCheck at 10 h predicted long term GFR decline more efficiently than sCr (R2 0.33 vs. 0.04).

The same predictive model has also been applied for AKI prediction. In this study, NephroCheck measured at 10 h along with kineticGFR measured at 4 h postoperatively were able to identify AKI as efficiently as sCr (i.e., the gold standard currently applied for postoperative AKI diagnosis). Nevertheless, this model predicted AKI more promptly compared to sCr. Combining these two markers in clinical practice would allow to stratify the risk of and to diagnose AKI for each patient as early as 10 h after surgery (at least 10–15 h before sCr). This time is usually crucial for physicians in adopting nephroprotective strategies and optimize perioperative care. Biomarkers of kidney insults might be used, beside sCr, for a more accurate and complete prediction of AKI and long-term GFR decline.

Interestingly, the adopted predictive models include biomarkers of both kidney dysfunction (kineticGFR) and kidney damage (NephroCheck). A recent important Consensus Statement [27] suggested that a combination of damage and functional biomarkers, along with clinical information, should be used to identify high-risk patient groups (such as in the post-surgery setting), improve the diagnostic accuracy of AKI, improve processes of care, and assist the management of AKI.

Moreover, preoperative serum NGAL and albuminuria (Table 1) might be considered in a panel of markers able to discriminate those patients predisposed to develop postoperative AKI and possibly be treated with less aggressive surgery.

Main strengths of the present study are the prospective design, the robust definition of AKI (based not only on sCr levels but also on urinary output), standardized anaesthesiologic protocols (Appendix A), and the comprehensive panel of biomarkers assessed, targeting different areas of the nephron at several time points after PN. Moreover, the long-term eGFR decline, which represents the primary outcome of this study, was assessed at 24 months after surgery. We also introduced for the first time the concept of subclinical AKI in Urology, comparing different biomarkers in predicting eGFR decline in a long-term follow-up, and validating their different cut-off for AKI.

The current study was not devoid of limitations. First, only 38 patients, in a single-centre study were analyzed; the results should require a validation in a multicentre setting with a larger cohort. Secondary, these biomarkers still have to prove their clinical effectiveness prior to their implementation into urologic surgery settings. Future studies are needed to demonstrate the cost-effectiveness of novel kidney biomarkers for the diagnosis of AKI in patients after renal surgery compared with current diagnostic method (monitoring of sCr level) and, more importantly, the impact of these tests on patient management and outcomes.

## 5. Conclusions

Blood and urine biomarkers measured peri-operatively may help to identify patients at higher risk for adverse long-term kidney outcomes. This represents, to the best of our knowledge, the first study on NephroCheck and kineticGFR in urology setting, supporting their potential role in patients who underwent NSS. Being readily available in routine clinical practice, NephroCheck and kineticGFR appear promising for early (10 h) or even very early (4 h) prediction of AKI and long-term GFR decline after PN. In patients undergoing NSS, NephroCheck and kineticGFR may complement sCr measurement or most likely improve the diagnostic accuracy.

## Figures and Tables

**Figure 1 biomedicines-11-01046-f001:**
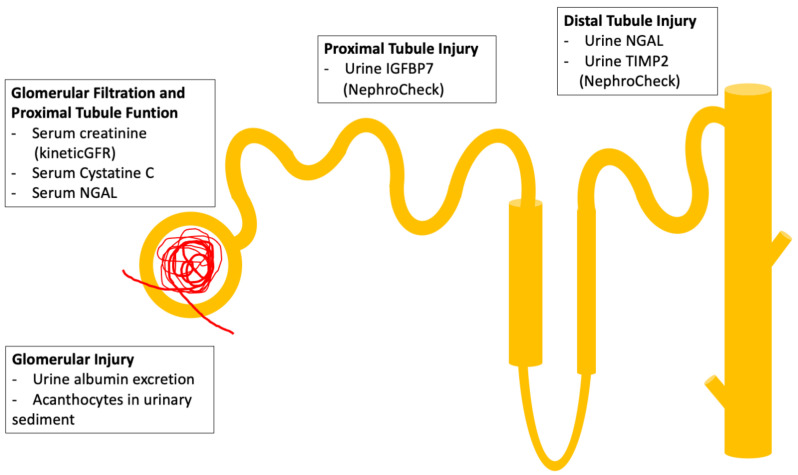
Schematic representation of different biomarkers of AKI which were selected and tested in our study in relation of sites of injury in the nephron. NGAL, neutrophil gelatinase-associated lipocalin; IGFBP7, insulin-like growth factor binding protein 7; TIMP2, tissue inhibitor of metalloproteinase-2.

**Figure 2 biomedicines-11-01046-f002:**
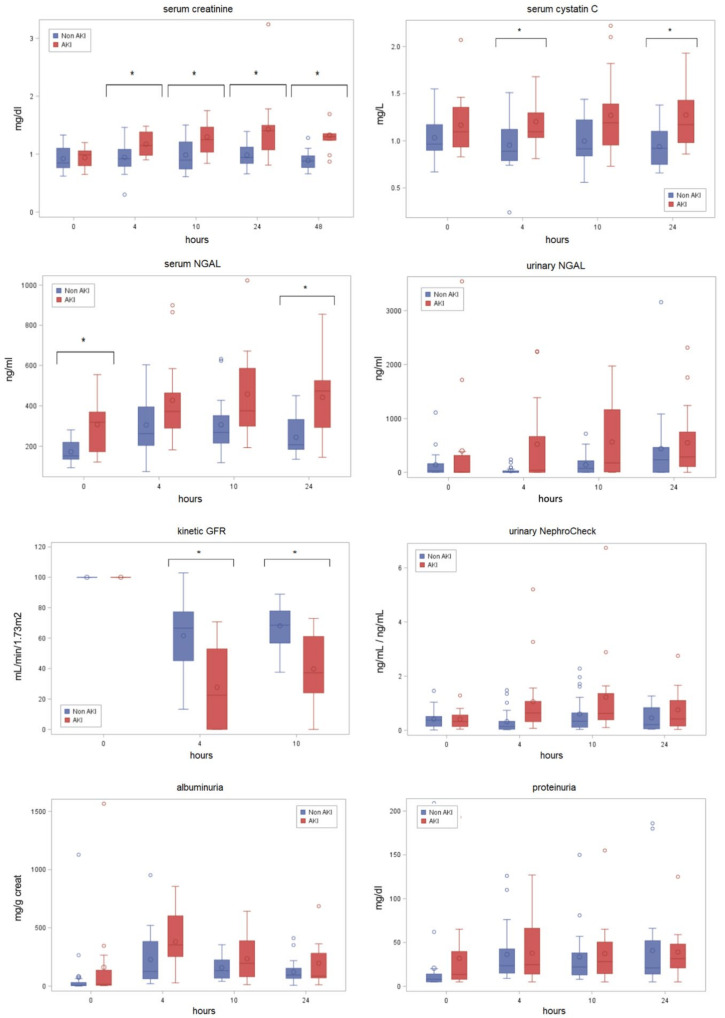
The postoperative change in biomarker levels from baseline to 24 h after surgery. Different graphs summarize the pattern of post-operative changes of the studied biomarkers as well as serum creatinine, both in patients who experienced AKI (red) and non-AKI (blue) Asterisk indicates *p* < 0.05. AKI; Acute Kidney Injury. Serum creatinine is expressed in mg/dL, serum cystatin C in mg/L, serum and urinary NGAL (neutrophil gelatinase-associated lipocalin) in ng/mL, kinetic GFR (glomerular filtration ratio) in mL/min/1.73 m^2^, Nephrocheck^TM^ is the ratio between TIMP-2 (tissue inhibitor of metalloproteinase-2) in ng/mL and IGFBP7 (insulin-like growth factor binding protein 7) in ng/mL and it is expressed as an AKIRISK^TM^ Score (≤0.3 low-risk of AKI, >0.3 high-risk of AKI), albuminuria is expressed in mg/g, proteinuria is expressed in mg/dL.

**Figure 3 biomedicines-11-01046-f003:**
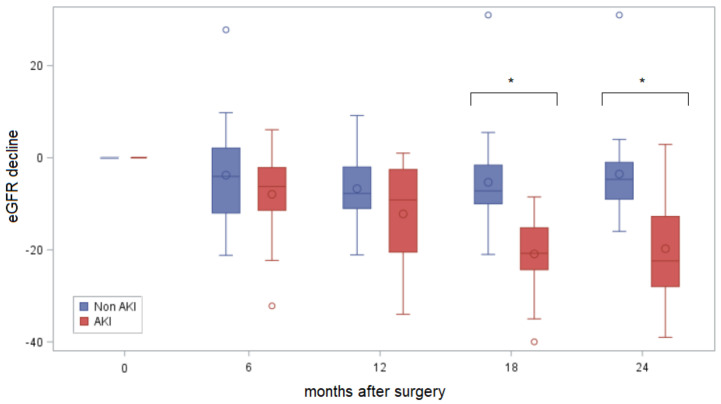
Estimated glomerular filtration rate (eGFR) decline at 6, 12, 18 and 24 months after surgery. * The median eGFR decline at 18 and 24 months after surgery was significantly faster in patients who experienced postoperative AKI. AKI; Acute Kidney Injury. eGFR in mL/min/1.73 m^2^, estimated using CKD-EPI formula.

**Table 1 biomedicines-11-01046-t001:** Preoperative characteristics of the patients enrolled in the study, according to postoperative AKI development.

	Tot	AKI	Non-AKI	*p*-Value
Patients, *n* (%)	38	16 (42%)	22 (58%)	
Age (years)	69.5 (62.0; 75.5)	68.5 (62; 75.7)	70 (62; 75.5)	0.76
Male gender, *n* (%)	27 (71)	13 (81.3)	14 (63.6)	0.30
**Preoperative comorbidities**				
BMI	26.3 (24.6; 29.4)	27.3 (25.3; 29.3)	26 (23.9; 29.4)	0.98
CKD, *n* (%)	10 (26)	5 (31.3)	5 (22.7)	0.83
Diabetes, *n* (%)	6 (16)	2 (12.5)	4 (18.2)	1.0
Heart disease, *n* (%)	11 (29)	3 (18.8)	8 (36.4)	0.30
Hypertension, *n* (%)	22 (58)	10 (62.5)	12 (54.5)	0.52
**Preoperative laboratory values**				
Hemoglobin (g/dL)	14.5 (13.6; 15.5)	14.6 (13.6; 15.8)	14.5 (13.6; 15.5)	0.69
Serum creatinine (mg/dL)	0.88 (0.78; 1.06)	0.99 (0.8; 1.06)	0.85 (0.77; 1.1)	0.79
Preoperative eGFR (mL/min/1.73 m^2^)	78.6 (67.4; 85.0)	72 (68; 84.6)	79 (64; 86)	0.94
**Preoperative biomarker values**				
Cystatin C (mg/L)	0.98 (0.92; 1.26)	1.10 (0.94; 1.36)	0.97 (0.90; 1.17)	0.14
Serum NGAL (ng/mL)	180.8 (142.6; 299.5)	319.5 (173.5; 370)	152 (136; 219.5)	0.002
Urinary NGAL (ng/mL)	4.0 (0.0; 227.9)	0.0 (0.0; 314.7)	25.2 (0.0; 161.3)	0.87
NephroCheck^®^ Test	0.35 (0.15; 0.53)	0.33 (0.14; 0.56)	0.38 (0.15; 0.51)	0.98
Proteinuria (mg/dL)	10.5 (5; 16)	13.5 (8; 39.5)	8 (5; 14)	0.09
Proteinuria/Creatininuria	0.13 (0.08; 0.26)	0.15 (0.10; 0.35)	0.11 (0.06; 0.22)	0.21
Albuminuria (mg/g)	9.3 (4.9; 63)	16.3 (6.7; 135.4)	7.5 (3.7; 30.4)	0.08
**Surgical parameters**				
Tumor size (cm)	3.5 (2.8; 4.8)	3.1 (2.4; 4.3)	3.8 (2.8; 4.8)	0.32
PADUA score	8 (7; 10)	8.5 (7; 10.5)	8 (7; 9)	0.53
Warm ischemia time (min)	12.5 (0; 20)	13.5 (0; 21)	12.5 (0; 20)	0.80
Operative time (min)	170 (130; 195)	175 (135; 208)	170 (120; 190)	0.76
**Outcome**				
12-month eGFR decline	−8.0 (−14; −2.1)	−9.15 (−20.5; −2.6)	−7.75 (−11; −2)	0.07
24-month eGFR decline	−11.5 (−19.5; −5.0)	−20.75 (−24.3; −15.2)	−7.20 (−10; −1.6)	<0.0001

BMI, Body Mass Index, CKD, Chronic Kidney Disease, eGFR (estimated glomerular filtration ratio) using CKD-EPI formula and it is expressed in mL/min/1.73 m^2^, PADUA, Preoperative Aspects and Dimensions Used for an Anatomical (PADUA) classification of renal tumors in patients who are candidates for nephron-sparing surgery. In accordance with KDIGO guidelines, CKD was defined as GFRcr < 60 mL/min/1.73 m^2^ and/or persistent proteinuria.

**Table 2 biomedicines-11-01046-t002:** Sensitivity and specificity of different promising biomarkers in early predicting postoperative AKI.

	4 h after Surgery	10 h after Surgery
Serum creatinine increase > 0.3 sensitivity specificity	25% ----	38% ----
Urinary NephroCheck > 0.3 sensitivity specificity	81% 73%	88% 50%
kineticGFR reduction ≥ 50% sensitivity specificity	56% 82%	56% 100%
Serum Cystatin C increase ≥ 10% sensitivity specificity	31% 96%	50% 96%

## Data Availability

The datasets used and/or analyzed during the current study are available from the corresponding author on reasonable request.

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
