# Peer review of "Novel Biomarkers for Early Detection of Acute Kidney Injury and Prediction of Long-Term Kidney Function Decline after Partial Nephrectomy"

_biomedicines, 2023, doi:10.3390/biomedicines11041046_

Round 1

Reviewer 1 Report

Authors measured several serum and urinary biomarkers in 38 RCC patients who received robotic NSS, and compared these markers to the risk and incidence of postoperative AKI and eGFR drop after one year. In an aggressive way, nephrololgists see patients with AKI experiences as CKD patients, even after clinical recovery. Therefore, the topic is worth serious digging. Comments are as the followings.

1. One of the major drawback is the small patient number. Readers will have doubt if the statistics might be under-powered. Please try to provide sensitivity and specificity, even though with such a small patients number. 

2. It is very inappropriate to use those 105 non-kidney tumor patients as the "matched control" and compare their kidney outcome. I strongly recommend not to do that.

3. Please revise the language part of your manuscript. For example, in your Results, "KineticGFR at 4 hrs (p=0.008) and urinary NephroCheck at 10 hrs (p=0.001) resulted 233 the biomarkers able to predict eGFR decline most efficiently at 24 months postoperatively." is unsound as an English sentence. As a reader, I surely understand what authors were referring to, but please have somebody (good in English) read the entire manuscript and pick up those sentences and revise them. 

4. The layout of your references numbering is not correct.

5. Readers will look forward to seeing the real world application of your own conclusion to justify the value.

Author Response

1. One of the major drawback is the small patient number. Readers will have doubt if the statistics might be under-powered. Please try to provide sensitivity and specificity, even though with such a small patients number.

Thank you for your comment. Planning a study on several new biomarkers is expensive, especially when multiple assessments at different time-points are needed to understand the best time point for each biomarker. This is mainly the reason why we recruited, as a pilot study, a small patient number. After we have identified the best biomarkers and their optimal time points in this specific setting, we have organized a multicenter study to prove their role in a bigger cohort of patients, but this study will take months to years more.

As suggested by the reviewer, we provided sensitivity and specificity for each promising biomarker in Table 2.

2. It is very inappropriate to use those 105 non-kidney tumor patients as the "matched control" and compare their kidney outcome. I strongly recommend not to do that.

The comparison between eGFR decline at 24 months after surgery of both groups of PN patients (with and without AKI) and the matched non-neoplastic control group (-2.7 ml/min) highlights the effect of renal mass reduction and the additive effect of AKI on long-term renal function. We thought this could have been an interesting message. However, as suggested by the reviewer, we removed this part from our manuscript.

3. Please revise the language part of your manuscript. For example, in your Results, "KineticGFR at 4 hrs (p=0.008) and urinary NephroCheck at 10 hrs (p=0.001) resulted 233 the biomarkers able to predict eGFR decline most efficiently at 24 months postoperatively." is unsound as an English sentence. As a reader, I surely understand what authors were referring to, but please have somebody (good in English) read the entire manuscript and pick up those sentences and revise them.

We have corrected the reported sentence. Moreover, we enlisted a native English speaker to help with rewriting.

4. The layout of your references numbering is not correct.

We have now accurately provided the references in the specific format requested by the journal.

5. Readers will look forward to seeing the real world application of your own conclusion to justify the value

Thank you for your kind words.

Reviewer 2 Report

This manuscript elegantly describes how novel renal biomarkers can predict AKI and serve as prognostic parameters. The study was supervised by experts in the field of AKI, and the data shed light on how such novel indicators can help avoid potentially nephrotoxic agents, especially in clinical situations such as surgery on the kidney itself. It sounds logical that proteins derived from tubular epithelial cells are rapid indicators of renal cell damage and have predictive value on eGFR decline later on.

Unexpectedly in this study, a decline of GFR is demonstrated in patients who undergo minimally invasive surgery and tumor resection by rNSS. A nephrologist could hypothesize that after 24 months the remaining nephrons would compensate for the loss of the resected part of the kidney. The warm ischemia time could be a potential reason for setting long-lasting injuries.

My main suggestion is that the sub-images of the combined Figure 1 should be shown larger and the parameter and unit of measurement should be indicated on the Y axis.

Figure 1 was published in the Journal of Urology: It is unusual to present in a research article a figure that has already been published in another journal. It must be modified or omitted.

Antonelli A, Allinovi M, Cocci A, et al. The Predictive Role of Biomarkers for the Detection of Acute Kidney Injury After 390 Partial or Radical Nephrectomy: A Systematic Review of the Literature

The discussion is long and does not embed these markers into the large array of markers that have been identified in the last few years such as IL18, Neprilysin, Proenkephalin……..

Minor points

Line 233: the wording sounds questionable. Suggestion: “resulted to represent the biomarkers”

Line 237 the wording also needs to be improved.

Line 287 "been" needs to be eliminated

Supplem. Figure 2 Title in part A needs rewording, Legend: misprint in Cystatin

Line 232 the word Table 2 misses “supplementary”

Author Response

This manuscript elegantly describes how novel renal biomarkers can predict AKI and serve as prognostic parameters. The study was supervised by experts in the field of AKI, and the data shed light on how such novel indicators can help avoid potentially nephrotoxic agents, especially in clinical situations such as surgery on the kidney itself. It sounds logical that proteins derived from tubular epithelial cells are rapid indicators of renal cell damage and have predictive value on eGFR decline later on. Unexpectedly in this study, a decline of GFR is demonstrated in patients who undergo minimally invasive surgery and tumor resection by rNSS. A nephrologist could hypothesize that after 24 months the remaining nephrons would compensate for the loss of the resected part of the kidney. The warm ischemia time could be a potential reason for setting long-lasting injuries.

Thank you for your kind words.

My main suggestion is that the sub-images of the combined Figure 2 should be shown larger and the parameter and unit of measurement should be indicated on the Y axis.

We modified the figure 2, accordingly.

Figure 1 was published in the Journal of Urology: It is unusual to present in a research article a figure that has already been published in another journal. It must be modified or omitted.

We are sorry for this mistake. We modified the figure 1.

 The discussion is long and does not embed these markers into the large array of markers that have been identified in the last few years such as IL18, Neprilysin, Proenkephalin…….. 

 A lot of different biomarkers have been proposed in the scientific literature in recent years, however only a little part has been tested in patients who underwent a partial nephrectomy.

Minor points

Line 233: the wording sounds questionable. Suggestion: “resulted to represent the biomarkers”

Line 237 the wording also needs to be improved.

We have corrected the reported sentences. Moreover, we enlisted a native English speaker to help with rewriting.

Line 287 "been" needs to be eliminated

We have corrected the reported sentences. 

Supplem. Figure 2 Title in part A needs rewording, Legend: misprint in Cystatin

Thank you for your suggestion. We corrected the mistake.

Line 232 the word Table 2 misses “supplementary”

Thank you for your suggestion. We corrected the mistake.

Round 2

Reviewer 1 Report

Authors have responded to comments. No more comment.